# Learning from a Feasibility Trial of a Simple Intervention: Is Research a Barrier to Service Delivery, or is Service Delivery a Barrier to Research?

**DOI:** 10.3390/healthcare8010053

**Published:** 2020-03-03

**Authors:** Julia Frost, Nicky Britten

**Affiliations:** St Luke’s Campus, University of Exeter Medical School, Exeter EX1 2LU, UK; n.britten@exeter.ac.uk

**Keywords:** applied health services research, competing organizational roles, high quality research

## Abstract

(1) Background: Applied health services research (AHSR) relies upon coordination across multiple organizational boundaries. Our aim was to understand how competing organizational and professional goals enhance or impede the conduct of high quality AHSR. (2) Methods: A qualitative study was conducted in two local health care systems in the UK, linked to a feasibility trial of a clinic-based intervention in secondary care. Data collection involved 24 semi-structured interviews with research managers, clinical research staff, health professionals, and patients. (3) Results: This study required a dynamic network of interactions between heterogeneous health and social care stakeholders, each characterized by differing ways of organizing activities which constitute their core functions; cultures of collaboration and interaction and understanding of what research involves and how it contributes to patient care. These interrelated factors compounded the occupational and organizational boundaries that hindered communication and coordination. (4) Conclusions: Despite the strategic development of multiple organizations to foster inter-professional collaboration, the competing goals of research and clinical practice can impede the conduct of high quality AHSR. To remedy this requires the alignment and streamlining of organizational goals, so that all agencies involved in AHSR develop a shared understanding and mutual respect for the progress of evidence-based medicine and the complex and often nuanced environments in which it is created and practiced.

## 1. Introduction

Applied health services research (AHSR) is essential to the progress of evidence-based medicine, but the relationship between the process of clinical research and healthcare outcomes is often confusing, with the wider benefits of research-active or research-intensive healthcare systems poorly understood [1]. AHSR typically occurs within complex systems, involving a large number of dynamic interactions between a diverse range of professionals working across occupational and organizational boundaries. This has led to calls for researchers to better understand the needs of health care decision makers, for example, in relation to the competing timelines of clinical research and practice and the contextual realities of delivery systems [2]. Here, we outline the key organizations in the UK and their objectives, some of which are comparable to health care systems elsewhere. 

The National Health Service (NHS) was created to provide comprehensive health care in England and Wales by an Act of Parliament in 1946. One of the seven core principles underpinning the current NHS Constitution is an aspiration for the highest standards of excellence and professionalism; including a commitment to the “conduct and use of research to improve the current and future health and care of the population” [3]. 

NHS Foundation Trusts (NHSFT) were established in 2004, with the objective of providing care to patients according to NHS quality standards and principles; but unlike existing NHS hospitals, they are accountable to local people, who can become members and governors; have freedom to decide locally how to meet their obligations; and are authorized and monitored by an independent regulator for NHS foundation trusts [4]. Within each NHSFT, Research and Development (R&D) Departments oversee research governance. These departments may act as the sponsor of research and provide oversight of research by guiding the Principal Investigators and managing the risks associated with any research initiated. They may also act as a host organization, whereby they set up externally sponsored research and support the Principal Investigators participating in those studies. Both sponsors and hosts may provide research sites for which R&D departments are responsible [5]. 

The UK Government’s revised mandate to NHS England for 2018-2019 [6] requires NHS England to promote and support participation by NHS organizations, patients, and carers in research funded both by commercial and non-commercial organizations, so that the NHS supports and harnesses the best research and innovations and becomes the research partner of choice. However, it is difficult to evaluate the impact of research activity within the NHS, because NHS Trusts do not aggregate these data due to resource constraints, while commercial companies do not share data because of commercial sensitivities [7]. Furthermore, there are tensions between the modernization of health services and the demands of AHSR. Research conducted in Canada found that while researchers and practitioners identify with the principles of collaborative research, the low value attributed to participation in research made it difficult for practitioners to collaborate [8]. Not all health professionals can choose to be involved in research or not; they may lack knowledge, skills, and confidence; have difficulties accessing support or mentorship; and lack supportive leadership [9]. 

Within the NHS, the Clinical Research Network (CRN) provides the infrastructure to support high quality clinical research studies. The CRN is comprised of both local networks and 30 clinical specialties, who coordinate and support the delivery of high-quality research. With national leadership and coordination provided by the CRN Coordinating Centre, the CRN is thought to support approximately 70% of all clinical research [7]. It has been estimated that in 2014/15, the CRN supported clinical research activities, which generated £2.4 billion of gross value added and almost 39,500 jobs [7].

There are thirty-three medical schools within the UK that are recognized by the General Medical Council (GMC), who regulate the profession at a national level (similar to Australia, New Zealand, and South Africa). Medical School staff and students also conduct a range of substantive and methodologically diverse research and, while research skills are a required outcome for medical graduates [10], they are rarely taught in a comprehensive and comparable way, nor are students’ attitudes to research routinely assessed [11]. 

Within UK medical schools, Collaborations for Leadership in Applied Health Research and Care (CLAHRCs) and Clinical Trials Units (CTUs) have played a key role in the design and conduct of AHSR. In the UK, thirteen CLAHRCs were funded by the National Institute for Health Research (NIHR) to undertake high-quality AHSR focusing on the needs of patients and support the translation of research evidence into practice within the wider NHS [12]. As with initiatives in Canada and the United States [13,14], one approach adopted by the CLAHRCS involved research users, including patients and clinicians, actively being involved in the prioritization of research topics, as a corrective to supply driven model of research development and conduct, often instigated by policy makers [2]. Nine NIHR Applied Research Collaborations (ARCs) began to replace the CLAHRCs in the UK from October 2019.

Like CLAHRCS, CTUs are multi-disciplinary specialist units that offer expertise across all required disciplines to ensure robust and effective clinical research conduct, but their emphasis is upon high quality clinical trials. In the UK, CTUs are based within universities and have had to comply with the European Directive for Clinical Trials. The involvement of a CTU in the planning and implementation of clinical research is recommended by funders (e.g., the NIHR) because they offer methodological knowledge and expertise, as well as the infrastructure, regulatory understanding, and experience in the practical management of clinical trials research [15]. However, research in both the USA and UK has demonstrated that CTUs have markedly different histories and infrastructure, which impact on the ways that they manage clinical trials [16,17]. 

Trust between the professionals and organizations involved in clinical research can be hindered by competition for increasingly scarce financial resources [18]; which can further be challenged by the lengthy and complex structures and processes of research governance and approvals [19]. The Health Research Authority (HRA) was established in 2011 to streamline the regulation of health and social care research in the UK, with the core purpose of protecting and promoting the interests of patients and the public in health and social care research. A key function of the HRA is the ethical review and approval of AHSR proposals. Critics suggest that members of research ethics committees (RECs) often review scientific aspects of research proposals, rather than their ethical implications [20], and that the increasing level of bureaucratization may undermine the competitive advantage of a publicly funded, open access health system for undertaking AHSR [21]. 

The aim of this paper is to use one particular research study to help understand how competing organizational goals enhance or impede the conduct of high-quality applied health services research. Given the potential sensitivities, we have not identified the study, the HRA REC that provided approval for the study, or cited its published outputs.

## 2. Materials and Methods 

### 2.1. Study Design

A qualitative study embedded in a pragmatic feasibility randomized controlled trial.

The research question for this study was generated from a research prioritization exercise, undertaken by a CLAHRC with active involvement from a patient involvement group and two secondary care physicians. The aim of the randomized controlled trial (RCT) was to test the feasibility of running a definitive trial of a particular clinic-based intervention. The study sample comprised of people with a particular chronic illness who were due to attend an outpatient clinic at one of two secondary care centers. We planned for a research nurse at each location to identify potential participants from the clinic lists of participating physicians. Support was sought from the local CRN, and the research was funded by the NIHR. The feasibility trial was conducted with support from the local CTU and the local Specialty Research Network (SRN). Potentially eligible patients who were due to attend a clinic appointment were identified from clinic lists by SRN nurses. Patients who were willing to be contacted were sent an information sheet about the trial and were contacted by phone by the nurse after one week, to confirm eligibility and discuss the study. Patients who wanted to take part in the study were then sent a consent form and baseline questionnaire. On receipt of the signed consent form, the patient was randomized. Participants were randomized in a 1:1 ratio to the intervention or control arms using a computer-generated random allocation sequence prepared by the CTU. An automated web-based system was used to conceal the allocation. Randomization was stratified by clinic session using randomly permuted block sizes in a non-systematic sequence. Because patients were added to the clinic list later than anticipated, multiple amendments to REC approval were required to facilitate recruitment. Participating patients were asked to arrive early at the clinic so that the intervention could be demonstrated to them (at the intervention sessions) before their consultations. 

### 2.2. Data Collection

We aimed to audio-record intervention sessions and clinical consultations across both trial arms and study sites, to understand the intervention mechanisms, and its subsequent utilization in practice, when compared with usual care (reported elsewhere). We also planned to conduct semi-structured interviews with a purposive sample of 5 clinical staff and 10 patients in both trial arms to explore wider organizational factors. All data were transcribed and anonymized and managed using Nvivo 11 software [22].

### 2.3. Data Analysis

We employed discourse analysis, a methodology that focuses upon the socio-cultural and political context in which text and talk occur, with the goal of identifying cultural hegemony and how it is (re)produced [23]. We used a loose set of questions to interrogate the data, focusing on: The assumptions that underpin what is being said; the discursive resources used to construct meaning; the potential consequences of the discourse; and what may be gained or lost as a result of these deployments [24].

## 3. Results

For the feasibility study, all physicians working at the two centers were approached. Nine consented to participate and were formally inducted into the trial by the CTU. Of nearly 400 patients screened for eligibility to participate, less than 75 were recruited to the study—roughly half each in the intervention and control arms, and significantly less than we had anticipated. 

In terms of the qualitative work, and with written consent, we audio recorded intervention sessions, clinical consultations, and conducted semi-structured interviews with a purposive sample of 12 staff (comprising six physicians, two CRN managers, two clinical nursing staff, and two research nursing staff), and 12 patients (six in each trial arm). We interviewed more staff and patients than we had originally planned (24 rather than 15) because of the difficulties encountered in the conduct of the feasibility study and in order to explore wider organizational factors. Study participants were given pseudonyms. 

Our analysis identified differences in stakeholders’ distinct modes of engaging with research; their cultural norms, values, and preferences for engaging with research; and their particular ways of understanding research; and framed our understanding of the interface of clinical and AHSR at multiple organizational levels. Figure 1 illustrates the configuration of clinical and research organizations at both sites. 

### 3.1. Physicians’ Reactions

The enthusiasm of the two physician co-applicants was not always shared by their clinical colleagues and was tempered by both the complexity of the trial processes and the low rate of study recruitment. 

At an early Trial Management Group (TMG) meeting:
“[Dr Abbot] suggested that participation rates are likely to be high and estimated an attrition rate of 50% for the 6-month follow up….” (Minutes of TMG Meeting, 2 years pre-trial)

However, the final patient recruitment rate was less than 25% in both centers, and this co-applicant shared their disappointment after the trial had concluded:
“I was quite disappointed that we didn’t get the recruitment as we needed, because I think initially I thought that oh, this would be plain sailing, and so many obstacles came around… it’s a learning experience.” Dr Abbott

The other physician co-applicant suggested that perhaps the difficulties were clinical and, that with hindsight, the research should have been conducted in a different setting or in a different patient population, although this was discussed at length during the planning process:
“So I think there would be more interest in doing this in people at or soon after diagnosis… that would mean a primary care study… if you could do it outside the consultation room it’s probably, yeah, maybe it could be simplified.”Dr Blake

In Center 1, one of the participating physicians told us that they had planned to implement a similar intervention to that being trialed, and also that they practiced in a way that prioritized time management:
“My initial perception was that I like to think that I conduct … consultations vaguely in that manner anyway…the times where I say ‘what do you want to talk about’, tend to be those that I know the consultation will last for an hour if I don’t limit it to that.” Dr Cross

In contrast, physicians in Center 2 valued the opportunity that the intervention afforded to practice in a way that was more holistic, with the objective of improving the patient’s outcomes, rather than managing time:
“One of the reasons I was interested in the study was ‘cause I am concerned, so—my background to this is I watch my friends who’ve all become GPs, get trained in consultation skills and how to set up consultations… and I’m very conscious that we, as hospital physicians, have actually no such training… So being involved with something that was actually looking at the consultation itself, trying to structure it and measuring effect and seeing whether the outcome was more satisfactory for patients was appealing.” Dr Dicker

Across both centers, some physicians declined to participate, while others were formally inducted, however, counter to the research protocol, their use of the intervention and/or recording of clinical consultations was unreliable. Furthermore, when invited to participate, one physician asked to be a co-author on study publications, despite not meeting the criteria for authorship. Thus, a range of perspectives on research activity were identified:
“As in anything there are more doctors that are more research-friendly than others. So um, [Physician] is fantastic because he’s research based, he deals with a lot of research and, you know, he understands the process of research and maybe not all of the physicians are as keyed up in research as [Physician]…”Nurse Elion

### 3.2. Nursing Staff Reactions

Whilst the participation of the physicians was universally described as voluntary, both of the clinically-based nursing staff told us that they had not been given a choice about participation in the trial. In Center 2, this was because of their perceived capacity and familiarity with Information technology (IT):
“[Department manager] at the time approached me and asked if I would mind taking part in the study, because I used to do the [condition] clinics a lot and also because I was quite good at IT and I didn’t have any other roles within the department at the time so… She told me it would be on an iPad and you just had to press a couple of buttons and that was it….”Nurse Fleming

Alternatively, in Center 1, participation in the trial was despite their lack of IT or research experience:
“I thought ‘Ooh, what have I got myself into!’, um, not because of the study itself but basically some of the skills needed for the study. And I was thinking ‘Oh heck! I’m terrible with computers’ etc. etc. but in the end I just took it in my stride and thought ‘Well, yes, I can do this!’”Nurse Gatson

Despite the nurse in Center 2 having relevant skills for facilitating the intervention, they felt as though other clinical staff resented them having a new hybrid research role, and subsequently resigned their clinical post:
“Purely because in their opinion I was a [clinical nurse], so, bottom of the picking litter [sic]. And actually if they want jobs done, I’m the one that, they want me to go and do it and couldn’t appreciate – it was almost like it would have run better if I was in a different hospital, if I was suddenly not around my colleagues, ‘cause they weren’t accommodating, they didn’t help. I was seen as actually just sitting in a room, ‘Oh why do you get to do that, why can’t we do that?’ Almost like school children being jealous that I’m getting almost a day off, when I wasn’t getting a day off, I was just helping take part in something else that they’d already said they didn’t want to do or they weren’t suitable to help with.”Nurse Fleming

In addition to the clinically-based nurses, two research nurses were provided by the CRN to work across the interface of research and clinical practice. The duration of the trial was extended to maximize recruitment, which meant that the CRN nurses became further involved in facilitating the research.

In Center 1, the CRN nurse was critical of the departmental manager for not booking patients into clinics and not providing a room for the study in a timely manner, but was unaware that a particular research room in the department was only provided by CRN involvement:
“I think the main problem was getting the clinic dates and information from [Departmental manager]…she’s not clinically-based or, you know, she wasn’t clinical, she was an admin person. I don’t think she understood research… as with any research that we take part in, we have to ring the patient, get the verbal consent and then send them the information within a certain length of time so then they can decide whether they want to take part in the study or not. And then, for us, we would do that, and then they would contact [Hospital] and then [CTU] had to get more information out to the patient and then get it back… All they have to do is, you know, that day you just book an extra room, and I don’t understand why that was a problem. When I went we went in the room upstairs.”Nurse Elion

This CRN nurse used her experiential knowledge to target specific patients for recruitment:
“I’d look at the list and I’d think ‘I know them… they’ll do it.’ So that was quite useful… And I think if you recognize patients’ names and patients have taken part in other research studies and have said they’re happy to take part in research. Then I think it’s only right to include those patients, ‘cause you think ‘Oh yeah, they will take part’”Nurse Elion

The CRN nurse in Center 2 employed a similar approach, but was critical of the research administrators, who were perceived as only working in the research department because it was a ‘soft option’:
“Um [the screening team] made one phone call and if they didn’t get that person that seemed to be enough work done, um, with regard to that patient, whereas I thought we needed to be chasing them up and one phone call just wasn’t enough… I’d already done a study similar to this, so I already know that you can’t ring someone up and expect an answer there and then, you have to keep chasing… I just thought one phone call and a tick in a box to say ‘I’ve done that job’, wasn’t good enough….”Nurse Hackett

As part of the study induction, both clinical and research nurses participated in an induction day which involved recruitment training provided by patient representatives. However, both of the CRN nurses were disparaging of the patient representatives’ enthusiasm, which they believed did not match the likely telephone responses of ‘real patients’ to recruitment:
“You had some of the public members in… I think the people that you had in were eager, and I think that’s that side of the spectrum, but there wasn’t anybody, even if it was like an actor, with a negative, um, spectrum, do you understand what I mean?... people that were ringing patients asking them if they’d like to take part… maybe, they wouldn’t have known what to have said to somebody who was like ‘No, why? duh!’”Nurse Elion

### 3.3. Network Managers’ Reactions 

In Center 1, the CRN manager was critical of the research team for not working more collaboratively with the CRN during the planning of the trial:
“I was approached by one of the SRN nurses to look at an early form of protocol, so I kind of saw it in its very early stages... And then when it finally came on to the books it was probably about a year later, and we were involved, not so much in the feasibility but once it had been accepted at the site and they realized that they needed some involvement from the research team, so slightly the wrong way round.”Nurse Inglis

However, the research team had made several attempts to approach and work with the CRN during the planning of the research, including a presentation about the proposed design at a CRN meeting:
“[CRN]: Negative feedback received; the SRN indicated at their recent meeting that their primary interest is in projects with potential for income generation. [researcher] has responded to SRN email re areas of clarification but has not had a response yet.”(Minutes of TMG Meeting, 1 year pre-trial)

Attempts were also made by patient representatives to keep the planning of the trial on the CRN agenda:
“[Patient] reported that the [CRN] lay panel had met and requested that the [Trial name] study will be a rolling agenda item. [Patient] will inform the panel that we will be happy to talk to them once the study has progressed.”(Minutes of TMG Meeting, 1 year pre-trial)

The CRN manager at Center 1 was also critical of the clinical co-applicants and researchers for having unrealistic expectations of the rates of patient recruitment to the feasibility study (a key objective of which was to establish whether this kind of project was feasible), which required additional resourcing which had not been planned for or costed:
“Um, it was easy to identify the patients, but there weren’t enough of them in the clinics that were identified, but then it involved actually speaking to them… so it meant doing some evening calls, which is not unusual practice, but we felt that that was necessary in order to get hold of people. Um, and then we had to talk to them about it, send out information, they had to post back to [CTU], they then posted stuff out to them, and this all had to happen in quite a quick, tight, timeframe, which often meant that patients weren’t then being seen in the clinic as research patients, they kind of missed the timings and missed the opportunity.” Nurse Inglis

In Contrast, the CRN manager at Center 2 was critical of clinical colleagues, who were perceived as not acknowledging the role of research in NHS business:
“The problem’s with [Department]! I would say I don’t just get this with this study, I get it with a lot of studies, it’s like: actually just do it, it’s part of NHS core business doing research and it’s actually very ‘them and us’, and that’s the thing throughout the - not throughout the whole of the NHS, but we get that quite a lot in studies ‘Oh that’s research, that’s not us’, but it is actually you, because we are the NHS and we are research as a core business, etc.…And it’s just one of those things, until there’s a change of culture within the NHS, it’s not going to change.”Nurse Joshi

### 3.4. Patients’ Reactions 

Trial recruitment and participation rates were lower than expected in both centers. While Centre 1 had a higher recruitment rate, it was noted that patients often declined to participate because they already had resources that the intervention was designed to facilitate:
“Some of the people that had [condition] for a long time ‘It’s a bit bloody late for that’ or ‘If I don’t know what I’m doing by now …’ type thing”Nurse Elion

In contrast, recruitment was lower at Centre 2, where participants were more likely to struggle to engage with the intervention because of their poor literacy, which meant that more nursing time was utilized:
“One guy I had to read it for him because he couldn’t read, so he was just discussing the whole thing with me.”Nurse Fleming

When patients from Center 2 were interviewed about their experience of participating in the study, some struggled to discriminate between the impact of the trial processes (e.g., baseline and subsequent questionnaire booklets) and the actual intervention:
P:First [questionnaire is] obviously a bit daunting, ‘cause it’s, you see all these boxes, you think ‘Oh my God all these pages!’ But once you get into it, and I suppose ‘cause I’m used to doing them anyway…I would say, about an hour over all… ‘cause I took me time on it, I didn’t …I:Did the intervention in any way help [you]?P:I think [questionnaires] helped me more… Completing those give a chance to actually say what I thought of it, where I was going.Mr Smith

### 3.5. Resource Constraints 

Both co-applicant physicians reflected that what had been initially perceived of as a ‘simple intervention’ became more complex as it became ‘trialised’ and subject to regulatory and research processes:
“I think when the study arose it was a simple question and, I think, but when people talked about it and developed into a study, I think it became more complex, it was no longer a simple question. I think the question was still simple, but the process was [laughs] pretty long and complex… I thought that the patient would just come to the clinic, will [engage with the intervention] and will go home, but I think they have lots to do at home …in the process of the study”Dr Abbott

Despite there being no statistically significant difference in the length of the consultations between the intervention and control arms of the trial, several of the physicians in Centre 1 described how the intervention made their clinics longer, which led to penalties from the Departmental Manager:
“[Intervention] made the consultations longer …and bearing in mind we are very pressurized now, and we’re constantly pressurized that we’ve got to see more patients, and this economic clime that we are currently living in– I like motivational interviewing…[but] my consultations often go over and then I keep patients waiting and then I get management on my back…”Dr Lang

This was in contrast to some of the physicians in Centre 2 who described how the intervention made the consultation more focused and, in some cases, shorter:
I:OK. Um, what, if any, impact do you think [intervention] had…P:Possibly speeded it up…Because there was a focus… prioritizing from the patient’s perspective, um, that we could then, you know, set to on what they really wanted to talk about.Dr Ross

In terms of wider resource issues, physicians in both centers identified that the intervention would be difficult to implement in clinical practice without further development:
“It’s a research grant and so I think if you were going to put this into practice, I think, if you were going to make it viable, then there may have to be a change… that could make it more efficient and maybe that could be, um, telephone, you could actually do it by FaceTime, there may be other ways of doing it.”Dr Lang

Resource issues were also of concern to the CRN, who suggested that they had been financially penalized by the NIHR, as a result of not recruiting to the initial study target:
“The fact that we missed our [recruitment] target had implications for funding … which was really disappointing…so I think that felt really unfair, um, knowing how hard we’d worked here, you know, with [Nurse Elion] and the other [nurses], getting them on board and trained and research ready. And the amount of things we’d done kind of above and beyond, you know, the evening phone calling, setting up the pre-screening, um, logs so that we could then go back and cross-reference. I think that felt a bit sort of unfair, somehow.” Nurse Inglis.

## 4. Discussion

The study illuminated an array of organizational and occupational boundaries that hindered coordination and communication vital for the success of the research (Table 1).

It has wide ranging resource implications for our clinical and research partners. The conclusion of the study was that it was not feasible to move to a definitive trial of this ‘simple’ intervention. One might speculate that better organizational coordination would have led to a different result.

In Center 1, physicians highlighted their expertise in the delivery of health care, which was driven by operational concerns about time management, and where academic research was perceived of as hindering clinical practice. In addition, research staff in Center 2 identified that behavior of some clinical colleagues (e.g., disparaging comments) could hinder good research practice. International policy increasingly identifies clinical research as ‘core business’ for health services worldwide [8,19]; however, clinical staff are often required to recruit patients to clinical trials about which they may feel ambivalent or lack specialist training or leadership [25,26]. In contrast, a key enabler to clinical staff participation is the organizational culture, such as professional networks that champion research and leadership within organizations, which can model and normalize clinical staff participation in research [9]. More needs to be done to support and reward clinical staff participation in AHSR [27]. 

In Center 2, clinical staff embraced the opportunity to participate in AHSR, which they perceived could augment their clinical practice. A wide scale review of the international literature identified that participation by clinicians and healthcare organizations in research can improve healthcare performance, but that despite the proliferation of boundary spanning organizations and professional roles, the role of organizational determinants of implementation effectiveness remain unclear [28]. 

Research staff in Center 1 emphasized their expertise in conducting AHSR and reported that both academics and managers lacked insight into research conduct. This was despite both academic and patient partners having approached the CRN early in the development of the proposal, and our research meeting the criteria for high priority portfolio adoption [6]. The Department of Health has recently re-emphasized that ‘when resources are stretched it is important that NIHR CRN effort on studies with the highest priority is not diminished’ [6,10]; however non-clinical researchers must also be aware of the contextual realities of research business [2], which may include fiscal penalties. 

Despite evidence that patient involvement enhances patient recruitment to trials [29], and extensive support for patient involvement from CLAHRC infrastructure, patient recruitment to this trial was poor. The fact that patients struggled to engage with the research for clinical and administrative reasons and struggled to discriminate between trial processes and the actual intervention under study, suggests that further research is required to identify how best to minimize bureaucracy associated with ‘simple’ interventions and optimize the impact of patient involvement [29,30]. 

A strength of the study is the triangulation of physician, manager, nursing, and patient perspectives across organizational and professional settings, which enabled us to explore factors that can contribute to and hinder the conduct of AHSR. We agree with Boaz et al. [27] that, at organizational level, one measure of research engagement is the extent of patient enrolment in trials, but that other organizations factors, such as cross boundary communication, are also indicative of the wider research engagement culture. At an individual level, we were also able to identify individual attitudes that informed staff engagement, such as personal interest in the topic or an expectation that research was a key part of their role [26]. 

This paper is based on a single research study, and a further limitation was the number of participants who agreed to be interviewed. Our understanding would have benefitted from the perspectives of physicians who declined to participate in the research or those who used the intervention in a discretionary way, rather than as intended by randomization. Previous research has identified that clinical staff are often ambivalent about research processes such as randomization [25], and that this is associated with poor information provision to patients, and an inclination to view the results of the results as unscientific or irrelevant to clinical practice [31]. 

Thus, our findings demonstrate that despite the strategic development of multiple organizations to foster inter-professional collaboration (such as the CLAHRC or CRN), the competing goals of research and clinical practice can impede the conduct of high-quality AHSR [32].

## 5. Conclusions

This qualitative study enabled us to identify discrepancies between the assumptions that underpin discourse; the discursive resources used to construct meaning; potential consequences of the discourse; and the impact of these deployments for the key agents and organizations involved in a feasibility trial. We found different assumptions about the role of research in the NHS and discursive resources about the nature and value of ‘evidence’. We also identified strict adherence to policies and procedures which we believe were disproportionate to any risks associated with this feasibility study of a non-invasive intervention. 

In 2006, the Department for Health set out with a vision to improve the health and wealth of the nation through research, which aimed to: “Create a health research system in which the NHS supports outstanding individuals, working in world-class facilities, conducting leading-edge research, focused on the needs of patients and the public” [33]. We conclude that to fulfil this vision requires the alignment and streamlining of organizational goals, so that all agencies involved in applied health services research develop a shared understanding and mutual respect for the progress of evidence-based medicine and the complex and often nuanced environments in which it is created and practiced. This vision must convey the rationale for applied health services research and its potential benefits for the NHS and the patients that it serves. To do this, the HRA, NIHR, and NHS must target their own constituent organizations and partners, as much as politicians and policy makers.

## Figures and Tables

**Figure 1 healthcare-08-00053-f001:**
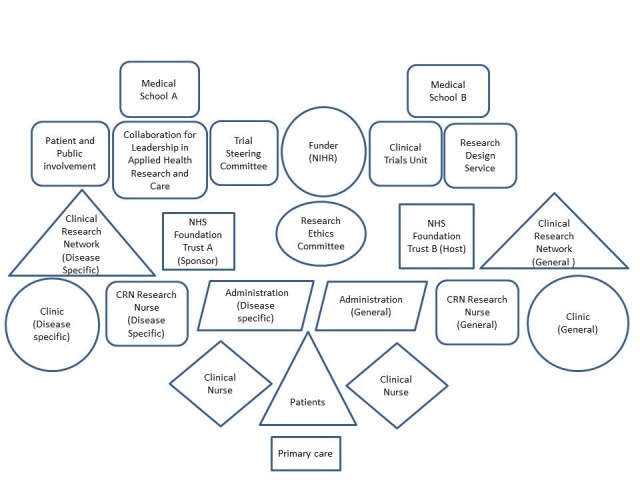
Organogram of clinical and research organizations at both sites.

**Table 1 healthcare-08-00053-t001:** Discourses employed by stakeholders in different roles.

Stakeholders	Assumptions Underpinning Discourse	Discursive Resources	Potential Consequences	What May Be Gained or Lost
Physicians	Co-applicants thought that recruitment would be easy. Participating physicians engaged with the research on their own terms.	Research processes disrupted research ethos. The research disrupted clinical practice, or physicians disrupted the research.	Perception of the trialization of something simple. Buy in: More holistic practice. Business as usual Subversion of trial protocol.	Disappointement and possible aversion to future research. Opportunity to empower individual patients. Lack of evidence to support the conduct of a full trial.
Nurses	Clinical nurses lacked agency. Research nurses were frustrated by boundary work.	New role was challenging. Extended role reinforced assumptions about research/practice gap.	Felt unsupported. Research/practice gap further substantiated.	Negative experience of research. Further evidence to support notion of research/ practice gap.
Network managers	Experiental knowledge of tensions between research and clinical practice	‘They’ (researchers and/or practioners) don’t know what they are doing.	Managers’ expertise is not recognised. Sense that no-one consults them or listens to their opinion.	Missed opportunities for shared learning or strengthening collaboration.
Patients	Like physicians, patients engaged with the research on their own terms.	Already have enough knowledge. Lack of understanding of the research.	Need for the intervention not well established. Patients confused by the trial processes.	Intervention may have been targeted in wrong population. Recruitment over complicated.
Shared perceptions about resources	Study perceived as overcomplicated.	Misperception that consultations took longer than usual.	Perception that intervention too difficult to implement in clinical practice.	Perception that research diffcult to conduct/ replicate in ‘real world’. CRN financially penalised.

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
