# Peer review of "Learning from a Feasibility Trial of a Simple Intervention: Is Research a Barrier to Service Delivery, or is Service Delivery a Barrier to Research?"

_healthcare, 2020, doi:10.3390/healthcare8010053_

Round 1

Reviewer 1 Report

None

Reviewer 2 Report

Thank you very much for this paper.

This manuscript is a resubmission of an earlier submission. The following is a list of the peer review reports and author responses from that submission.

Round 1

Reviewer 1 Report

I love the title and premise for this paper. The scene setting and overall presentation of the paper is really excellent and clear. However, I don't feel that the paper in its current form provides much in the way of novel contribution to the field. It is good to see further evidence drawn from a process evaluation to support themes in the existing literature on the complex interactions between research systems and healthcare. I would encourage the authors to dig a little deeper in terms of their analysis.  I was really looking forward to seeing what discourse analysis might deliver in terms of new insights as it is so rarely used in applied health services research. A couple of further comments:

- Now that ARCs have begun it would be helpful to add in a brief reference to them when you talk about CLAHRCS on P2

- It would be good to spell out what you mean by an 'exemplar study' - I hadn't come across this description before

- Is it necessary to include some of the methods for the trial itself (e.g. p3 lines120-130 and again in the results)

Reviewer 2 Report

Dear authors,

I had the great pleasure of reviewing the manuscript. First of all I would like to thank you for this unusual but wonderful paper. It addresses a totally important issue in healthcare research.

I wouldn't have a problem recommending the manuscript for publication in its present form, but I would like to draw your attention to 1-3 points I have truncated in advance in case you want to intervene.

In the introduction, you address the fact that the HRA is responsible for the ethical assessment of AHSR projects. Was this also the case with your study? I cannot find a separate hint to an ethics vote for the present research in the text (especially for the qualitative work). This could be completed or if this is not necessary in your system, I would be happy to receive the justification.

The methods section is very brief and I also asked myself in the inclusion of methodological aspects at the beginning of the results chapter what is now a feasibility intervention study and what is the qualitative work. In point 2.2 of the data collection, I asked myself whether the researchers were also questioned. With the persons listed here ("a purposive sample of 5 clinical staff and 10 patients") I get a problem with the participants at the beginning of the results chapter (6 physicians, 2 CRN managers,
and 2 clinical nursing staff, 2 research nursing staff, and 12 patients"). Perhaps you could make this a little clearer for your readers and thus easier to understand?

During the reading I was looking forward to the conclusions all the time. Precisely because the topic is so relevant, it would be really inspiring for us readers and researchers to read where the journey with this insight can or should go. Do you still have insights that you can share with the readers? This seems to me almost an immoral question, but I see this manuscript as an opportunity for health services research.

Thank you again for this paper and the opportunity to review it.

Sincerely

Astrid